# Do Amino Acid Antiporters Have Asymmetric Substrate Specificity?

**DOI:** 10.3390/biom13020301

**Published:** 2023-02-06

**Authors:** Gregory Gauthier-Coles, Stephen J. Fairweather, Angelika Bröer, Stefan Bröer

**Affiliations:** Research School of Biology, Australian National University, Canberra, ACT 0200, Australia

**Keywords:** LAT1, LAT2, y^+^LAT1, y^+^LAT2, asc-1, ASCT1, ASCT2, xCT, solute carrier

## Abstract

Amino acid antiporters mediate the 1:1 exchange of groups of amino acids. Whether substrate specificity can be different for the inward and outward facing conformation has not been investigated systematically, although examples of asymmetric transport have been reported. Here we used LC–MS to detect the movement of ^12^C- and ^13^C-labelled amino acid mixtures across the plasma membrane of *Xenopus laevis* oocytes expressing a variety of amino acid antiporters. Differences of substrate specificity between transporter paralogs were readily observed using this method. Our results suggest that antiporters are largely symmetric, equalizing the pools of their substrate amino acids. Exceptions are the antiporters y^+^LAT1 and y^+^LAT2 where neutral amino acids are co-transported with Na^+^ ions, favouring their import. For the antiporters ASCT1 and ASCT2 glycine acted as a selective influx substrate, while proline was a selective influx substrate of ASCT1. These data show that antiporters can display non-canonical modes of transport.

## 1. Introduction

Mammalian amino acid antiporters are an important element of amino acid homeostasis, allowing the exchange of amino acids between two compartments without compromising the chemical gradient of the complete amino acid pool. An elevated pool of all amino acids is maintained in most cells by transporters that accumulate a group of amino acids inside the cell either by Na^+^ cotransport or by electrogenic uniporters in the case of cationic amino acids. Functionally, transporters can thus be classified as loaders (symporters and electrogenic uniporters) and harmonizers (antiporters) [1]. Loaders accumulate a subgroup of amino acids in the cytosol and these are used as exchange substrates to bring in other amino acids through antiporters, thereby generating a harmonized pool of amino acids. Uniporters are usually expressed at low levels to avoid equilibration between cytosolic and plasma amino acid pools. 

Amino acid antiporters are the dominant contributors to transport activity in cultured cells when uptake is measured using radiolabelled amino acids [2]. Antiporters for small neutral amino acids are found in the solute carrier family 1, namely ASCT1 (sodium-dependent alanine–serine–cysteine transporter 1, SLC1A4) and ASCT2 (sodium–dependent alanine–serine–cysteine transporter 2, SLC1A5) [3] and in solute carrier family 7, namely asc-1 (sodium-independent alanine–serine–cysteine transporter 1, SLC7A10) [4]. Antiporters for large neutral amino acids are also found in the SLC7 family, namely LAT1 (large neutral amino acid transporter 1, SLC7A5) and LAT2 (large neutral amino acid transporter 2, SLC7A8). The transporters y^+^LAT1 (cationic [y^+^] and large neutral amino acid transporter 1, SLC7A7), y^+^LAT2 (cationic [y^+^] and large neutral amino acid transporter 2, SLC7A6) and b^0,+^AT (SLC7A9, broad neutral and cationic amino acid transporter) exchange neutral and cationic amino acids [5,6]. Lastly, xCT is an exchanger for cystine and glutamate (SLC7A11) [7]. All SLC7 family transporters form heteromeric transporters with the trafficking subunit 4F2–heavy chain (SLC3A2) and are called light-chains because of their relative migration in SDS gels. The exception is b^0,+^AT, which forms a similar heterodimer with rBAT (SLC3A1) [8]. In *Xenopus laevis* oocytes, rBAT forms a heterodimer with an endogenous transporter that is very similar to mammalian b^0,+^AT [9]. 

Structurally and functionally, antiporters, symporters and uniporters are closely related [10]. They have a single amino acid binding site; access to which is alternately provided at the two sides of the membrane. This involves transition of the transporter through several conformations (open–outside, occluded, inside–open) [11]. The hallmark of an antiporter is that the conformational changes only occur in the presence of a substrate. When only one central binding site is present, this automatically results in a 1:1 exchange stoichiometry. By contrast, return of the empty transporter to the other side of the membrane, is an essential step for symporters and uniporters. The difference is not absolute as uniporters with a high propensity of exchange occur [12]. As another example, glucose transporter GLUT1 has been shown to switch between uniport and antiport mode depending on the energetic state of the cell [13]. Similarly, antiport stoichiometries other than 1:1 have been reported for certain substrates [14,15].

Computational simulation of cellular amino acid transport can replicate cytosolic amino acid concentrations as observed in cultured cells in vitro in a variety of media [2]. For these simulations, two assumptions were made: first, endofacial and exofacial substrate specificity of antiporters are identical, and second, while endofacial K_M_ values can be different from exofacial K_M_ values all substrate K_M_ values are scaled by the same factor. Typically, endofacial K_M_ values for the same amino acid are significantly higher than exofacial K_M_ values [16,17,18]. We refer to this canonical type of antiporter as symmetric. 

In contrast to these canonical characteristics, asymmetric antiport has been reported in the literature. For ASCT2 (SLC1A5) Pingitore et al. [19] showed that Gln, Ser, Asn and Thr could trans-stimulate the uptake of radiolabelled glutamine, while efflux of glutamine could be trans-stimulated by Gln, Ser, Asn, Thr, Ala, Cys, Val and Met. 

Asymmetric antiport was also observed in the case of LAT2, where the ratio of endofacial/exofacial K_M_ values was ≈180 for alanine and isoleucine, but only 0.6 for glycine [16]. Notably, different K_M_ values cannot be compensated by differences in the turnover rate of the transporter (K_cat_) as the transporter passes through the same occluded state in both directions. If a particular amino acid does not efficiently reduce the activation energy, the conformational transition is slower in both directions. Thus, although the V_max_ of glycine for LAT2 was only 21% of the V_max_ of leucine in the study above, it would still be a preferred efflux substrate. 

The reported asymmetrical behaviour creates a problem as it results in the generation of substrate gradients without an apparent energy input (Figure 1). When all endofacial K_M_ values are scaled by the same factor, no accumulation of amino acids is observed in any compartment (Figure 1a). Changing the scaling for just one amino acid results in amino acid imbalances (Figure 1b). 

An asymmetric antiporter thus resembles a Maxwell demon and would violate the second law of thermodynamics. In the original thought experiment, Maxwell conceived a two-compartment system of gases that contained a small hole at the partition, which could be opened and closed, thereby temporarily connecting compartments A and B. Both compartments had equal temperature. A gatekeeper (the Maxwell demon) watched individual molecules and only allowed the swifter gas molecules to pass from A to B, while letting slower molecules pass from B to A. As a result, the temperature of compartment B would rise without the need for a heat source. In rebuke, Leo Szilard and Leon Brillouin showed that the gathering of information required by the demon would offset the reduced entropy.

J. B. S. Haldane later proposed that enzymes have some properties resembling a biological Maxwell demon, without violating thermodynamic principles [20,21]. Haldane pointed out that enzymes possess structural information that can change the outcome of a reaction. For instance, if 99% of substrate A is converted into product B and only 1% into product C in the absence of an enzyme, adding an enzyme that catalyses conversion into C will transiently increase the amount of C at the expense of B. However, this information is applied in both directions and therefore does not change the equilibrium of the reaction [21]. Similarly, in the case of an antiporter the structural information content resulting in substrate specificity should apply in both directions. A mechanistic asymmetry would require energy input of some sort or a temporal change of mechanism.

To explain unusual behaviour of transport processes without violating the second law of thermodynamics, Lon Van Winkle proposed that cells continuously invest energy to maintain asymmetric membrane bilayers and to incorporate membrane proteins in a directional manner, which could provide an indirect source of energy to generate substrate gradients [22]. Similarly, transient storage of energy as torsion has been proposed as a mechanism to avoid energetic mismatches during the synthesis of ATP by ATP-synthase and the use of ATP by myosin [23].

To investigate whether antiporters can carry out asymmetric antiport we have developed a novel LC–MS method in which unlabelled amino acids in the oocytes’ cytosol exchange with ^13^C/^15^N amino acids in the medium. This allows the concurrent detection of amino acid transport in both directions from complex mixtures. 

## 2. Materials and Methods

RNA extraction and reverse transcription—The mRNA for asc-1 was isolated from human brain tissue, that of ASCT1 from 143B cells, and mRNAs for y^+^LAT1 and xCT were isolated from A549 lung adenocarcinoma cells. Cloning of human ASCT2, 4F2hc and y^+^LAT2, LAT1 and LAT2 was described previously [24,25,26]. RNA extraction was performed using the RNeasy Mini reverse transcription kit (Qiagen 74104), according to the manufacturer’s instructions. The resulting yield was determined using a SpectraMax QuickDrop (Molecular Devices). Two micrograms of total RNA were then reverse transcribed using SuperScript II reverse transcriptase (Invitrogen; 18064014) for a duration of two hours. Single stranded DNA were generally stored for a maximum of one day before being amplified by PCR in preparation for ligation into the pGHJ oocyte vector using Gibson cloning. 

Gibson cloning*—*To prepare coding DNA for ligation into the oocyte expression vector pGHJ [27], primers were designed to amplify the sequence with the addition of overlapping regions homologous to pGHJ cut at the *Eco*RI site (Table 1). The NEBuilder assembly tool (New England BioLabs) was used to this end. PCR was performed using *Pfu* (Promega; M7741) or *Taq* (Qiagen; 201203) DNA polymerases. PCR products were separated using gel electrophoresis and extracted using the Monarch DNA gel extraction kit (New England BioLabs; T1020S). The pGHJ oocyte vector was prepared for Gibson cloning through digestion of the *Eco*RI restriction site in the MCS using the *Eco*RI-HF restriction enzyme (New England BioLabs; R3101S) according to the vendor’s instructions. Ligation of the insert into pGHJ was conducted using the Gibson assembly cloning kit (New England BioLabs) according to the vendor’ instructions by combining 70 ng of linearised pGHJ vector with a two-fold molar excess of the insert fragment. NEB 5-alpha competent cells were transfected with the reaction product, allowed to recover and plated on ampicillin agar plates. Clones with plasmids of the expected size were propagated before isolating the plasmids using the Wizard Plus SV Miniprep DNA purification system (Promega; A1223). Plasmids were then sent to the Biomolecular Resource Facility (ANU) for Sanger sequencing. 

In vitro transcription*—*Two verified clones per transporter were selected for in vitro transcription and linearising the plasmids with *Sal*I-HF (New England BioLabs; R3138T) and purifying the product with the phenol chloroform method. The mMessage mMachine T7 transcription kit (Invitrogen) was used to transcribe capped RNA (cRNA) containing the coding sequence of the human amino acid exchanger, flanked by β-globin 5′ and 3′ UTR. Transcription occurred over two hours at 37 °C and RNA was extracted using the phenol chloroform method and ethanol precipitation. 

Frog surgery and oocyte preparation*—*Prior to microinjecting oocytes with cRNA, the ovaries of *X. laevis* frogs were harvested. Surgeries were conducted in accordance with the Australian code for the use of animals for scientific purposes and with approval from the ANU Animal Experimentation Ethics Committee (A2020/28). Once ovaries were extracted, they were placed in OR^2−^ (82.3 mM NaCl; 2.5 mM KCl; 1 mM MgCl_2_; 1 mM NaHPO_4_; 5 mM HEPES; pH 7.8) and cut up into smaller sections. Defolliculation was facilitated using collagenase B (Roche) at 0.6 mg/mL for approximately 12 h at 16*–*19 °C. Oocytes were then washed copiously with OR^2+^ (OR^2-^ with the addition of 1.8 mM CaCl_2_) and stored in the same buffer with the addition of gentamycin (50 μg/mL). Stage V oocytes were then selected and microinjected with 40 nL of water containing cRNA (5*–*10 ng h4F2hc; 5 ng hEAAT1; 10*–*15 ng hASCT1; 10 ng hASCT2; 5 ng hLAT1; 10 ng hLAT2; 10 ng hy^+^LAT1; 10 ng hy^+^LAT2; 10 ng hasc-1; and 10 ng hxCT). Expression was allowed to proceed generally over five days before flux experiments were conducted, except for LAT1 and LAT2, which were assayed three days post-injection. 

Radiolabelled flux experiments*—*For all radiolabelled transport experiments, oocytes in lots of 5*–*12 were dispensed into disposable transfer tubes. Oocytes were washed thrice with ND96 (96 mM NaCl; 1 mM MgCl_2_; 1.8 mM CaCl_2_; 5 mM HEPES; pH 7.4) and incubated with 100 μL of ND96 containing 100 μM of non-labelled substrate + ≥2000 cpm/nmol of radio-labelled substrate for the periods indicated in figure legends at room temperature. In some cases, an inhibitor or competitor was included, typically at a saturating concentration. To stop transport, oocytes were washed thrice with in ice-cold ND96 and individually dispensed into scintillation vials. For efflux experiments, the supernatant was sampled prior to this step. Oocytes were then lysed with the addition of 200 μL of 10% SDS and incubated for around 30 min. Three millilitres of Ultima Gold scintillation fluid (PerkinElmer) was then added before vials were capped, vortexed and placed either in a PerkinElmer TriCarb 4910 TR or Hidex 300 SL to measure scintillation. A volume equivalent to one nanomole from each transport buffer containing radio-labelled substrate was also sampled and counted as a stock reference before initiating the experiment. Counts from all samples were converted into transport rates using the following equation:$$\frac{cpm_{sample}}{cpm_{stock}}\;\times 1000\;=\;pmol/oocyte/24\;\min$$

Stable isotope flux tracing experiments*—*To measure rates of amino acid influx and efflux in a complex mixture, oocytes were aliquoted, ten at a time, into disposable transfer tubes, washed thrice with ND96 and incubated in a ND96-based solution containing all amino acids, except glutamate and aspartate, each at a concentration of 1 mM. The anionic amino acids were excluded from this formulation because they are already highly abundant in the cytosol of *X. laevis* oocytes. Cysteine was added to this formulation freshly to ensure minimal oxidation to cystine. After two hours of preincubation in this mixture, one set of oocytes was washed thrice with ND96 before snap-freezing in liquid nitrogen and storing on dry ice. The other sets of oocytes were washed thrice and then incubated in 100 μL of either plain ND96 or ND96 with all proteinogenic amino acids (each 0.1 mM) uniformly labelled with ^13^C and ^15^N (Cambridge Isotope Laboratories; MSK-CAA-1; pH 7.4). It should be noted that in this isotopically labelled mixture, cysteine exists in its oxidized form as cystine. The resulting trans-stimulation of transport activity was allowed to proceed for 30 min before 50 μL of the supernatant was sampled, after which oocytes were washed with ND96, snap-frozen with liquid nitrogen and stored on dry ice. In all cases, at the end of each series of washing, residual buffer was carefully removed using a pipette. 

For metabolite extraction, oocytes were collected from the transfer tubes by adding 600 μL of 60% (*v/v*) methanol and transferred to microcentrifuge tubes. To these tubes, 200 μL of chloroform was added and the oocytes were vortexed at 3200 rpm for five minutes to ensure efficient homogenisation, before centrifuging at top speed for a further five minutes. Either 20 μL, 200 μL or 300 μL of the aqueous fraction, depending on the analysis, was transferred to a clean microcentrifuge tube for drying in a vacuum concentrator. Supernatant samples were also de-solvated in the same manner in preparation for chemical derivatisation and LC–MS analysis.

LC–MS quantification of amino acids*—*Two methods were used for the quantification of amino acids from oocyte lysates and efflux samples. The main method involved butylating the amino acids and was able to quantify cystine and all proteinogenic amino acids except for cysteine. Poor detection of cysteine, likely due to its reactive thiol group has previously been reported [2]. Moreover, leucine and isoleucine could not be distinguished with this method due to identical retention times and similar fragmentation patterns. This method has previously been described with some modifications [28]. Briefly, dried extracts were solved in 120 μL of n-butanol:acetyl chloride (4:1), vortexed for five minutes and incubated at 65 °C for 25 min. Dried external standards were prepared in the same way and were sourced from a non-labelled amino acid mixture (Sigma; AAS18) supplemented with an equimolar amount of glutamine, asparagine, and tryptophan. Internal standards consisted of [^13^C-U] [^15^N-U] amino acids (Cambridge Isotope Laboratories; MSK-CAA-1) that were also processed in the same manner but were resuspended in n-butan-d_9_-ol: acetyl chloride (4:1). All samples and standards were dried in a vacuum concentrator for at least 3 h at 37 °C. Samples and standards were then dissolved in 10 mM ammonium acetate : acetonitrile (93:7; +0.15% formic acid). External standards were formulated with the following final concentrations: 0.05; 0.2; 1; 5; 10; 20; 50; 100; 200 μM. Both external standards and samples were spiked with internal standards at a final concentration of 5 μM and were vortexed, centrifuged at top speed for five minutes and transferred to LC–MS vials. Separation of analytes was achieved by a Kinetex 1.7 μm C18 100Å 100 × 2.1 mm column installed in an UltiMate 3000 UHPLC with a WPS-3000 Split Loop RS sampler and DGP-3600RS pump module (all from Thermo Fisher). This UHPLC was coupled to a Q-Exactive Plus mass spectrometer (Thermo Fisher). The injection volume was set to 4 μL, the flow rate to 300 μL/min and the column was maintained at 35 °C. Running solvent A was composed of 10 mM ammonium acetate + 0.15% formic acid and solvent B was acetonitrile +0.15% formic acid. All reagents and solvents used were LC–MS-grade. The runtime was 21.5 min, starting with a gradient elution of 7% solvent B held until the 4.0 min mark, after which B was linearly increased to 80% at 12.0 min. Solvent B was held at 80% until it was returned to 7% between 17.0 and 17.1 min. It was held there to re-equilibrate the column until the end of the run. Analytes were analysed in full scan mode using positive ionization. The mass filter was set to 100*–*400 m/z and resolution was set at 70,000 at 200 m/z. Automated gain control target was 5 × 10^6^ charges, maximum injection time of 150 ms and the number of micro-scans was restricted to one. Sheath gas flow was 48, auxiliary gas flow was 11, sweep gas flow was 2, spray voltage was 3.5 kV, capillary temperature was 256 °C and auxiliary gas temperature was 413 °C. Due to the need to quantify all three isotopomers per amino acid (non-labelled, labelled and internal standard), parallel/multiple reaction monitoring was not used as it would have overburdened the duty cycle, resulting in poorly defined chromatograms. 

For the detection of cysteine and its labelled isotope in ASCT1- and ASCT2-expressing oocytes, another chemical derivatisation method was used. Adapted from Sutton et al. [29], *N*-ethylmaleimide (NEM) was used to react with the sulfhydryl group of cysteine. Briefly, 20 μL of the aqueous phase from the extraction described above was transferred to a clean microcentrifuge tube and dried in a vacuum concentrator. To each tube containing dried sample, 6 μL of 10 mM NEM (dissolved in 10 mM ammonium acetate; pH 7.0) was added and the tubes were repeatedly brushed against a corrugated surface to help solve the metabolites. In parallel, external standards containing cysteine were prepared in the same manner. This was also true for the internal standard stock, except that it was reacted with *N*-ethyl-d_5_-maleimide (Cambridge Isotope Laboratories; DLM-6711-10). After ten minutes, the contents in each sample tube were collected at the bottom of the tube through centrifugation and 54 μL of acetonitrile containing the internal standards was added. The resulting mixture was briefly vortexed, centrifuged and transferred to a LC–MS vial. Internal standards were included at a final concentration of 5 μM across both samples and external standards. Cysteine separation and detection by LC–MS was performed similarly to the method described above except that a SeQuant ZIC cHILIC 3 μm 100Å 150 × 2.1 mm column (EMD Millipore) was used. Solvent A was held at 15% for the first two minutes of the run and then linearly increased to 85% by the 6.0 min mark. Solvent A was held until 11.5 min before it was returned to 15% within a span of 0.5 min. The total run time was 15 min and flow rate was 0.4 mL/min. The mass spectrometer ran in positive mode with a scan range of 120 to 650 m/z, a resolution of 70,000 at 200 m/z, an automated gain control target of 3 × 10^6^ charges and a maximum injection time of 50 ms for full scan mode. Parallel reaction monitoring for cysteine was also enabled and had a resolution of 17,500, an automated gain control target of 2 × 10^5^ charges, a maximum injection time of 50 ms and an isolation window of 0.8 m/z. Sheath gas flow was 50, auxiliary gas flow was 13, sweep gas flow was 3, spray voltage was 3.5 kV, capillary temperature was 263 °C and auxiliary gas temperature was 425 °C.

For both methods, quantification was achieved by integrating the area under the curve (AUC) for all analytes and their isotopomers and dividing by the AUC of their respective internal standard. These ratios were converted to concentrations using the calibration curve constructed from the external standards. Data processing was facilitated by Xcalibur software suite (Thermo Fisher) and Skyline (MacCoss Lab Software 22.2; [30]). Quantities were expressed as rates using the following formula:$$\frac{\left(\frac{{\left[AA\right]}_{vial}\;\times \;V_{vial}}{V_{dry}}\right)\times \;V_{fraction}}{N_{oocyte}}\;\;=\;pmol/oocyte/30\;\min$$
where *[AA]_vial_* and *V_vial_* are the concentration of a given amino acid in the LC–MS vial and the total volume solved, respectively; *V_dry_* is the volume of methanol extraction solution that was dried; *V_fraction_* is the fraction of the total methanol extraction represented by the volume taken for drying; and the denominator is the number of oocytes contained in each transfer tube (i.e., 10).

## 3. Results

### 3.1. Experimental Design

Oocytes have an endogenous amino acid pool containing millimolar concentrations of anionic amino acids, around 200–500 μM of cationic amino acids and < 100 μM of neutral amino acids [16,31]. To prime oocytes for the simultaneous determination of inward and outward fluxes, they were first incubated for 2 h in a 1 mM mix of unlabelled neutral and cationic amino acids (^12^C). This generates an elevated intracellular pool of neutral and cationic amino acids for subsequent exchange. 

Because of their natural abundance, anionic amino acids were excluded from the preloading mix. Oocytes were subsequently extracted to analyse the cytosolic amino acid pool (preload sample) (Figure 2a). Identically treated oocytes were then washed and incubated with 0.1 mM labelled amino acids (^13^C) in ND96 or in plain ND96 buffer for 30 min. A sample from the supernatant (supernatant sample) was taken before washing oocytes and subsequently extracting the cytosolic amino acid pool (extract sample). We detected comparable amino acid concentrations after preloading for 2 h with 1 mM ^12^C amino acids (Figure 2b) in oocytes expressing the antiporter trafficking subunit 4F2hc, or the heteromeric amino acid antiporters 4F2hc–LAT1 and 4F2hc–LAT2. The concentrations were similar to those reported previously [16,31]. Notably, Gly levels were the same between LAT1- and LAT2-expressing oocytes although different ratios of endofacial and exofacial K_M_ values were reported for LAT2 [16], demonstrating that LAT2 does not act in a Maxwell demon-like fashion. 

### 3.2. Flux Analysis of Antiporters

#### 3.2.1. LAT1 and LAT2

4F2hc–LAT1 and 4F2hc–LAT2 are well-characterized antiporters mediating a 1:1 exchange of their substrates [5,6]. However, there is evidence that LAT2 may carry out facilitated diffusion to some extent [32,33]. As derived from individual flux assays, LAT1 has a substrate specificity that comprises large non-polar amino acids (branched-chain and aromatic amino acids), while LAT2 accepts all neutral amino acids except proline. Because 4F2hc interacts with several oocyte endogenous light-chains [34], transport of amino acids was corrected by its activity in the absence of any light-chain. This allowed the calculation of net import ([^13^C] amino acids entering the oocyte) and of net export ([^12^C] amino acids exiting the oocyte). 

Our experiments confirmed the narrower substrate specificity of LAT1 compared to LAT2, which included bidirectional transport of Ile/Leu, Phe, Trp, Tyr and His (Figure 3a). Methionine was not a significant LAT1 import substrate but was exported, albeit weakly. The apparent opposite fluxes of cationic amino acids were most likely the result of an overcompensation when subtracting 4F2hc-mediated fluxes. When 4F2hc is expressed alone it interacts with an endogenous light-chain that is similar to y^+^LAT1, where cationic amino acids move in the opposite direction of neutral amino acids (see below). Overall, there was little evidence for asymmetric substrate specificity mediated by LAT1. 

Preferred substrates of LAT2 for import and efflux were Thr, Phe, Tyr, Trp, His and Asn. Smaller fluxes were detected for Ala, Val, Ile/Leu and Ser. Methionine has been shown to be a strong substrate when presented alone, but its flux was minimal in a complex mixture. Glutamate and aspartate are not substrates of LAT2 and instead were most likely formed from other amino acids by transamination [35]. The opposite movement of cationic amino acids was again most likely caused by overcompensation when subtracting the activity of 4F2hc-expressing oocytes. A striking asymmetry was observed for Gln, which was exported but not imported. This was surprising, as Gln is a known LAT2 substrate when presented individually [33]. 

To further investigate the potentially asymmetric transport of glutamine, we used radiolabelled glutamine in the presence of a mix of unlabelled amino acids (0.1 mM each, Figure 4a) to match the stable isotope experimental condition. As expected, unlabelled amino acids reduced glutamine uptake through competition, but residual import was significantly above background (Figure 4a). Glutamine was also a robust efflux substrate, which could be released after preloading through the addition of 1 mM unlabelled extracellular glutamine (Figure 4b). The reason for the discrepancy between the two analogous experiments remained unclear.

Although both transporters carry out a 1:1 exchange [16], import was calculated to be higher than export (Figure 3). The most likely explanation for this discrepancy was that exiting intracellular ^12^C amino acids experience competition from imported ^13^C amino acids. Vice versa, extracellular ^13^C amino acids do not experience competition because exported ^12^C amino acids were diluted into the much larger supernatant. As a result, we could only compare the substrate specificity of import and export, but not their quantity.

When expressed in oocytes, we detected no evidence for uniport via LAT1 or LAT2. In fact, oocytes expressing 4F2hc alone showed a higher level of leakage than oocytes expressing 4F2hc–LAT1 or 4F2hc–LAT2 (Figure 4c). 

#### 3.2.2. y^+^LAT1 and y^+^LAT2

The antiporters y^+^LAT1 and y^+^LAT2 also form heteromeric transporters with 4F2hc [8]. Mechanistically, they are different to the other antiporters as they are designed to carry out asymmetric transport. Both transporters accept cationic and neutral amino acids, but the K_M_ of neutral amino acids decreases by several orders of magnitude in the presence of Na^+^ [36]. As intracellular Na^+^ is low in oocytes [37], export is mainly in the form of cationic amino acids while import is predominantly neutral amino acids plus Na^+^. As a result, an asymmetry is introduced due to the prevailing Na^+^ concentrations not because of structurally encoded selection of substrates by the transporter. 

Our LC–MS analysis demonstrated that y^+^LAT1 facilitated a stricter vectorial transport than y^+^LAT2 (Figure 5). In both transporters cationic amino acids could serve as export or import substrates, while neutral amino acids showed a preference for import, which was particularly obvious for y^+^LAT1. Both transporters preferred large neutral amino acids, such as Leu, Ile, Met, Gln and Asn. The experiments could not discriminate whether Glu was an import substrate or was generated by transamination.

The substrate specificities of y^+^LAT1 and y^+^LAT2 were consistent with previous reports [26,38,39,40]. The antiporter y^+^LAT2 is widely distributed, while y^+^LAT1 is predominantly found in intestinal, renal epithelial and blood cells. Vectorial transport is essential for absorption and reabsorption of cationic amino acids, explaining the stronger discrimination between import and export substrates in the epithelial isoform y^+^LAT1. 

#### 3.2.3. asc-1 and xCT

The transporters asc-1 and xCT form heterodimers with trafficking subunit 4F2hc. The asc-1 transporter is expressed in brain and adipose tissue where it serves specific functions such as the transport of D-amino acids involved in neural signalling and adipocyte differentiation [41]. Substrates of the transporter were Gly, Ala, Ser and Thr (Figure 6). As observed for other transporters the subtraction of the transport activity of 4F2hc alone resulted in an overcompensation of arginine and lysine transport, resulting in apparent transport in the opposite direction. The transport activity of asc-1 was lower than that of the other transporters thereby decreasing the signal-to-noise ratio.

The physiological role of xCT is an exchange of intracellular glutamate for extracellular cystine. The transporter plays an important role in the maintenance of the redox balance in cells by providing cysteine for glutathione biosynthesis [42]. Glutamate and cystine (net cystine import = 1.3 ± 0.4 pmol, measured in a separate experiment) were the only import substrates of 4F2hc–xCT (Figure 6). Since intracellular cystine is reduced to cysteine inside the cytosol, glutamate is the only observable efflux substrate. Thus, the asymmetry of exchange is driven by metabolism not by the transport protein. In contrast to other glutamate transporters xCT discriminates against aspartate. 

#### 3.2.4. ASCT1 and ASCT2

The antiporters ASCT1 (SLC1A4) and ASCT2 (SLC1A5) are related to brain glutamate transporters [43]. While the canonical glutamate transporters are symporters, ASCT1 and ASCT2 are obligatory antiporters. Both require Na^+^ for functional antiport [44] but the sodium electrochemical gradient has no apparent effect on the directionality of the fluxes due to high-affinity binding of Na^+^ on both sides of the membrane [45]. Transporters of the SLC1 family form trimers, but do not require any trafficking subunits for surface expression. 

We observed that ASCT1 exchanged Ser, Thr, Asn and Ala, and Asp (Figure 7a), while ASCT2 exchanged Gln, Thr, Ala, Asn, and Ser (Figure 7b). Cysteine, which we analysed using a specific method [46], was not abundant in oocytes (≈167 μM). Fluxes in either direction were very small (<1 pmol/oocyte per 30 min). 

For ASCT1, proline and glycine were detected to be import- but not efflux-substrates. Glycine was also an asymmetric substrate for ASCT2. Aspartate was a substrate of ASCT1 but not of ASCT2. Although it had a large standard deviation, the efflux of aspartate was confirmed in a second series of experiments with a net export flux of 16 ± 18 pmol/oocyte. Although it is known that aspartate is a substrate of ASCT1 at low pH it is not considered a substrate at neutral pH [47]. To further investigate the asymmetric flux of proline in ASCT1 we used radiolabelled [^14^C]proline. Consistent with the LC–MS data we observed that proline was a weak substrate of ASCT1 but even in the presence of competing unlabelled amino acids a net influx remained (Figure 8a). Efflux of preloaded proline in exchange for 1 mM Thr by contrast, remained under the level observed in non-injected oocytes (Figure 8b).

Aspartate transport by ASCT1 has previously be shown to occur at pH 5.5 [47] but as demonstrated here also occurs at neutral pH in both directions. To confirm aspartate fluxes we measured uptake of extracellular aspartate (Figure 8c) and efflux (Figure 8d) of preloaded radiolabelled aspartate in exchange with 1 mM Ala in ASCT1-expressing oocytes. Both were significantly higher than those observed in non-injected oocytes. A similar observation was made for glutamate in the case of ASCT2 [48].

## 4. Discussion

Detailed mechanistic analysis of transport proteins over many years has firmly established that the accumulation of substrates is consistent with prevailing substrate and ion gradients; examples in references [49,50,51]. Except ASCT1 and ASCT2, all antiporters investigated in this study displayed canonical behaviour and symmetric exchange of amino acids. For ASCT1 and ASCT2, glycine and proline (ASCT1 only) were found to be import but not export substrates.

Structurally, three states of antiport proteins are relevant to substrate selectivity. The outside–open, inside–open and occluded state [10,52]. Transporters undergo an induced fit transition state, i.e., the transporter closes to form tight contacts with the substrate. The extracellular K_M_ is determined by the local conformation of the outside–open state and the intracellular K_M_ is determined by the local conformation of the inside–open state [18]. The occluded state is critical for substrate selection, because the tolerance for energetically unfavourable ligand–protein interactions is at its lowest. The contacts in this state are the same in both directions, explaining why substrate selectivity is normally symmetric in both directions. 

Notably, ASCT1 and 2 are structurally different from all antiporters in this study. ASCT1/2 have the glutamate transporter-fold (Glt), while SLC7 family transporters have the LeuT-fold [10,53]. In glutamate-type transporters a whole domain makes an elevator-like movement across the membrane after closure of a gate formed by hairpin-loop 2 [54]. Closure of the loop triggers translocation. In LeuT-fold transporters, by contrast, helix 1 and 6 perform a rocker-switch-type of motion, which is triggered by binding of the substrate. The rocker-switch motion allows alternate opening of the central binding site without vertical movement of a whole domain. It is conceivable that closure of the hairpin-loop 2 of ASCT1 and 2 is triggered even in the absence of substrate under certain conditions, allowing uncoupled transport to occur.

Two previous studies demonstrated the possibility that obligatory antiport can be uncoupled by certain substrates, thereby suggesting a mechanism for the observed asymmetries. Extracellular application of the amino acid analogue aminoisobutyric acid (AIB) to rBAT-expressing oocytes was reported to cause release of intracellular (trans) amino acids similar to alanine, while uptake was only 1/30 of that of alanine [14]. Thus, while Ala/Ala exchange was obligatory (1:1), Ala/AIB exchange was approximately 30:1. The same observations were made by Scalise et al. in the case of ASCT2. Here cysteine was a very poor uptake substrate, but caused the efficient release of preloaded radiolabelled glutamine in ASCT2-containing proteoliposomes and cells, similar to other substrate amino acids [15]. Translocation by ASCT2 was also investigated by Garaeva et al. [55], where different amino acids were preloaded into ASCT2-containing proteoliposomes, causing uptake of radiolabelled glutamine in the order Gln = Ala = Thr = Ser = Asn > Cys. A similar experiment in reverse direction was performed by Broer et al. [56], in which radiolabelled glutamine preloaded into oocytes was released by ASCT2 substrates in the order: Ser = Ala = Cys = Gln = Thr > Leu > Gly (Asn not tested). 

This behaviour could be explained by the binding to allosteric site(s), which facilitate transport of a trans-substrate without being translocated itself. Such a mechanism is related to that proposed for the symporter LeuT [57]. In this transporter the main substrate leucine is trapped in the binding site 1 (S1) until a second substrate binds to the allosteric site S2. This triggers release of Na^+^ and substrate from the S1 site into the cytosol. The S2 site was hence named the “symport-effector site”. Intriguingly, trapping is observed for leucine but not for alanine, although alanine can act as a symport-effector for leucine. In analogy an “antiport-effector site”, which can only be occupied by selected substrates (AIB for rBAT and cysteine or glycine for ASCT2), could trigger the release of a trans-substrate. The function of allosteric “antiport-effector sites” could be influenced by mutations of the transporter. For instance, mutation R365W in rBAT appears to selectively affect arginine efflux without altering arginine influx or the transport of leucine [58]. Notably, this mutation is located in the trafficking subunit, which has multiple contacts with the catalytic subunit. 

An alternative allosteric site was discovered by Garaeva et al. [52]. In the inward open cryo-EM structure of ASCT2, a lipid molecule, most likely phosphatidylcholine, was identified close to the substrate binding site. The head-group of the lipid was located where the gate-forming hairpin-loop 2 would be located in the inward occluded conformation. Several more lipid-binding sites were identified and classified as potential allosteric sites. Thus, the local lipid environment could influence the conformational changes occurring during transport. It is thought that ASCT1 and ASCT2 are forced into antiport because of high-affinity binding of Na^+^ on both sides of the membrane [45]. Sodium ions are actively translocated by ASCT2 [44,59] but the stoichiometry can differ between substrates [60]. The phospholipid choline-group may interfere with Na^+^-binding by the transporter. If a lipid would prevent access of Na^+^ and certain substrates, the transporter might be able to transition through the membrane without bound substrate. Proline is an unusual substrate of ASCT1 as it appears to be exchanged with limited accompanying exchange of Na^+^ [60]. The related glutamate transporters have three Na^+^-binding sites, one of which has to be occupied before the substrate can bind [54]; however, the corresponding sites in ASCT2 have not been reported. The mechanism of ASCT1 and ASCT2 is further complicated by the occurrence of an anion conductance [44,45,61] and partially voltage-dependent transport steps that involve binding of Na^+^ [45,62]. If glycine translocation could be accompanied by a different number of Na^+^ ions depending on the local concentration of lipids and Na^+^, directional transport could be readily explained. 

The best method to detect non-canonical activity of antiporters would be to inject into oocytes an amino acid mix that yields a final concentration of approximately 0.5 mM of all amino acids (except glutamate and aspartate). The oocytes can then be incubated in an equimolar amino acid mixture or ND96. In these experiments absolute quantification of all amino acids would be sufficient to detect any net fluxes. Although we did not harmonize intracellular amino acid concentrations, the lack of change of intracellular glycine in LAT2-expressing oocytes clearly demonstrated that the observed asymmetry of glycine affinity did not result in the depletion of glycine at the expense of other neutral amino acids, thereby excluding any Maxwell demon-like action. 

The use of mass spectrometry to analyse fluxes of all known transporter substrates at the same time is an attractive alternative to radioactive uptake assays with individual amino acids. We have previously used GC–MS in a qualitative way to analyse amino acid fluxes into and out of oocytes incubated in complex biological matrices [35]. These experiments confirmed earlier experiments using radiolabelled amino acids and transporter electrophysiology. Here we refined this method using LC–MS to quantify the fluxes of 20 amino acids. Cysteine was analysed separately after blocking its sulfhydryl group [46]. We further refined our previous method by incubating oocytes containing preloaded ^12^C amino acids with a pool of ^13^C/^15^N amino acids. Because of the mass difference this allows a separate quantification of amino acids moving from out-to-in (^13^C) from those moving in–to–out (^12^C). Although conceptually attractive, we encountered several problems. The first problem was that import and export do not seem to occur with a 1:1 stoichiometry. For several of the antiporters investigated here a 1:1 stoichiometry has clearly been established experimentally [16,26]. Moreover, efflux of intracellular amino acids strictly depends on the presence of extracellular substrates (see for example Figure 4b), a key feature of an obligatory antiporter. The only meaningful explanation for the discrepancy is a competition between incoming ^13^C amino acids and exiting ^12^C amino acids. Thus, some of the ^13^C amino acids return to the medium. This is likely because of the ultrastructure of the oocyte [63]. Amino acids will initially accumulate in the perimembraneous space before slowly diffusing further into the oocyte where the egg yolk is located. Evidence for this can be derived from experiments where transporter substrates were preloaded or injected. While preloaded radiolabelled amino acids can be fully recovered from oocytes by exchange, injected radiolabelled substrate can only partially be recovered (≈50%) [64]. Thus, it appears likely that ^13^C amino acids accumulate to higher concentrations in the vicinity of transporters, thereby effectively competing with the endogenous pool for release. Injecting high concentrations of unlabelled amino acids could also reduce the competition between imported ^13^C amino acids and cytosolic ^12^C amino acids. Another factor contributing to the discrepancy are the low medium concentrations of released amino acids. The average solute-accessible volume of a stage 6 oocyte is 364 ± 21 nl [64], while the supernatant was 100 µL, resulting in a >250-fold dilution of amino acids in the medium. The method and instrumentation used here provide calibration curves over several orders of magnitude but extracellular samples as a result had higher standard deviation than intracellular samples.

A second problem is the metabolic conversion of other amino acids into glutamate and aspartate. Alanine and aspartate transaminases are highly expressed in many cell types allowing the rapid conversion of a variety of amino acids into glutamate and aspartate, provided that sufficient oxaloacetate and α-ketoglutarate are available [65]. As a result, we have viewed changes of anionic amino acids as arising from metabolism and transport. 

Although there is far less endogenous amino acid transport in oocytes than in mammalian cells, it can be modulated by expression of heterologous mRNAs. This is particularly obvious for the 4F2hc trafficking subunit that increases the activity of an oocyte endogenous transporter with properties similar to y^+^LAT1. The co-expressed heterologous transporter light-chain does not fully outcompete the endogenous transporter and as a result we subtracted the activity of 4F2hc-injected oocytes, which, in turn, led to an overcompensation of fluxes, particularly for cationic amino acids. 

Despite these limitations, the method reliably identified transporter substrates in complex mixtures of amino acids that are largely in agreement with experiments using individual amino acids. A potential exception were sulphur-containing amino acids, for which fluxes were unusually small. For instance, methionine has been reported as a high-affinity substrate for LAT1 [66] and LAT2 [33], but the fluxes observed in this study were very small. The reason for this discrepancy remains unclear.

Transport in complex mixtures does provide a better insight into physiologically meaningful transport selectivity. For instance, previous characterization of ASCT2 suggested a significantly wider substrate specificity than ASCT1 [56,67], while transport in a complex mixture suggested that both transporters are quite similar with the main difference being that glutamine is a major substrate of ASCT2, while ASCT1 excludes it. This is in agreement with electrophysiological studies [45].

By contrast, y^+^LAT1 and y^+^LAT2 appear to be more different than previously thought. y^+^LAT1 has evolved to carry out vectorial transport of cationic amino acids in epithelial cells, while y^+^LAT2 is a more general amino acid exchanger that connects the pools of cationic and neutral amino acids. This is consistent with our previous observations that a combination of the cationic amino acid uniporter cat-1 and y^+^LAT2 can serve as a tertiary active transport mechanism to import neutral amino acids into cancer cells [2]. 

Our study shows that antiporters can have asymmetric substrate specificity, particularly when movement of ions is involved, as in the case of y^+^LAT1, y^+^LAT2, ASCT1 and ASCT2. Given that structural studies are now feasible for many of these transporters, it will be interesting to see whether certain substrates occupy allosteric sites and whether this affects the transport cycle. 

## Figures and Tables

**Figure 1 biomolecules-13-00301-f001:**
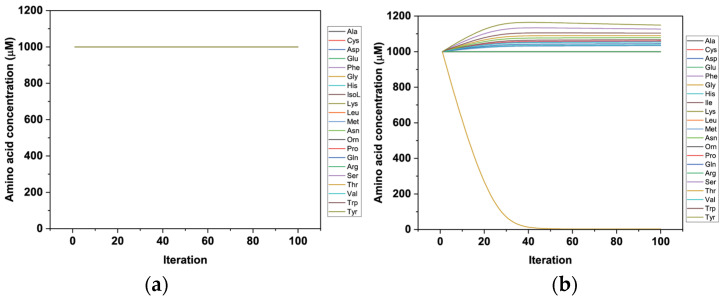
Simulation of symmetric and asymmetric antiporter kinetics in an artificial cell containing LAT2 only. The concentration of cytosolic and extracellular amino acids was set at 1 mM. (**a**) Intracellular concentrations of all substrate amino acids when LAT2 intracellular K_M_ values were scaled to be 180-fold higher than extracellular K_M_ values. (**b**) Intracellular concentrations of all substrate amino acids when the intracellular K_M_ value of Gly was scaled to be 0.6-fold higher than the extracellular K_M_ value (all other K_M_ values as in (**a**)). Gly (beige) is leaving the cell in exchange for all other substrates of the transporter.

**Figure 2 biomolecules-13-00301-f002:**
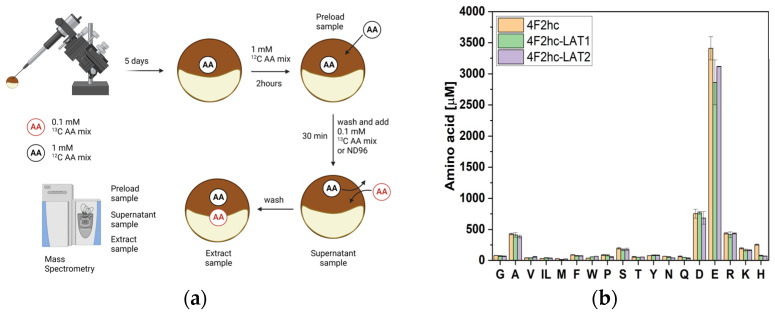
Analysis of *Xenopus laevis* oocyte amino acid pools. (**a**) After expression for five days oocytes were incubated with ^12^C (black) and ^13^C amino acids (red) in the order shown. Three samples were analysed for each experiment, namely the preload sample, supernatant sample and extract sample. (**b**) Cytosolic amino acid concentrations after incubation in a 1 mM ^12^C amino acid mix of oocytes expressing 4F2hc, 4F2hc + LAT1 or 4F2hc + LAT2 (Preload sample). Isoleucine and leucine cannot be separated by LC–MS and their sum is shown as IL.

**Figure 3 biomolecules-13-00301-f003:**
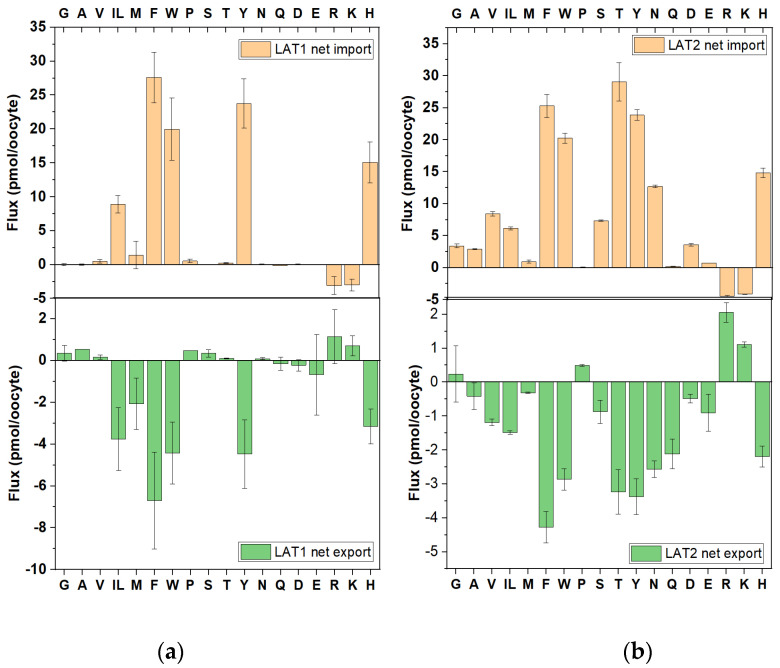
Analysis of antiport via LAT1 and LAT2**.** Oocytes expressing 4F2hc–LAT1 (**a**) or 4F2hc–LAT2 (**b**) were incubated with labelled and unlabelled amino acids as described in Figure 2a. Net import (positive numbers) and net export (negative numbers) are shown. Isoleucine and leucine cannot be separated by LC–MS and are shown as IL. The activity of oocytes expressing 4F2hc alone was subtracted, resulting in apparent oppositely-directed transport for some amino acids.

**Figure 4 biomolecules-13-00301-f004:**
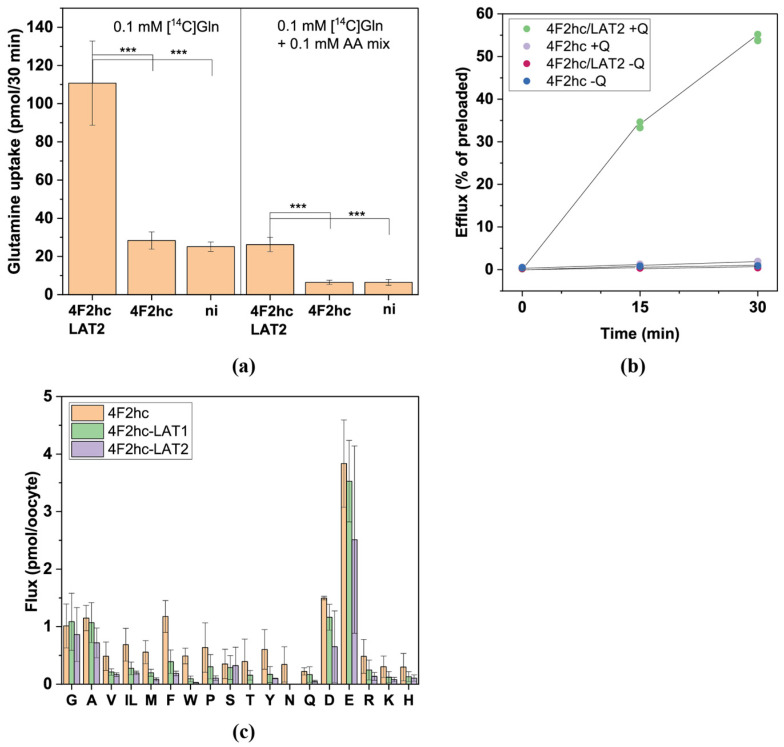
Analysis of asymmetry and uniport activity of LAT1 and LAT2. (**a**) Uptake of 0.1 mM [^14^C]glutamine was measured in the presence and absence of an unlabelled amino acid mix (0.1 mM each) in oocytes expressing 4F2hc, 4F2hc–LAT2 or non-injected oocytes. (**b**) Oocytes expressing 4F2hc, 4F2hc–LAT2 or non-injected oocytes were preincubated with 0.1 mM [^14^C]glutamine for 1h. After washing, oocytes were incubated in ND96 in the presence (+Q) and absence (−Q) of 1 mM glutamine and supernatant samples were analysed for radiolabelled glutamine after 15 and 30 min. (**c**) Oocytes expressing 4F2hc, 4F2hc–LAT1 or 4F2hc–LAT2 were incubated in ND96 for 2 h after which the supernatant was analysed for amino acids to calculate net efflux. Isoleucine and leucine cannot be separated by LC–MS and are shown as IL. *** Significant difference at *p* < 0.0001.

**Figure 5 biomolecules-13-00301-f005:**
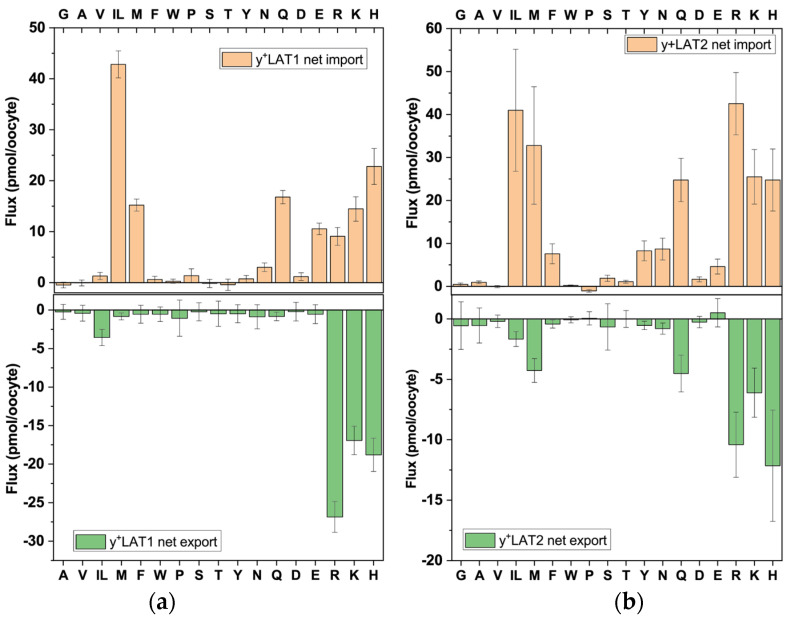
Analysis of antiport via y^+^LAT1 and y^+^LAT2**.** Oocytes expressing 4F2hc–y^+^LAT1 (**a**) or 4F2hc–y^+^LAT2 (**b**) were incubated with labelled and unlabelled amino acids as described in Figure 2a. Net import (positive numbers) and net export (negative numbers) are shown. Isoleucine and leucine cannot be separated by LC–MS and are shown as IL.

**Figure 6 biomolecules-13-00301-f006:**
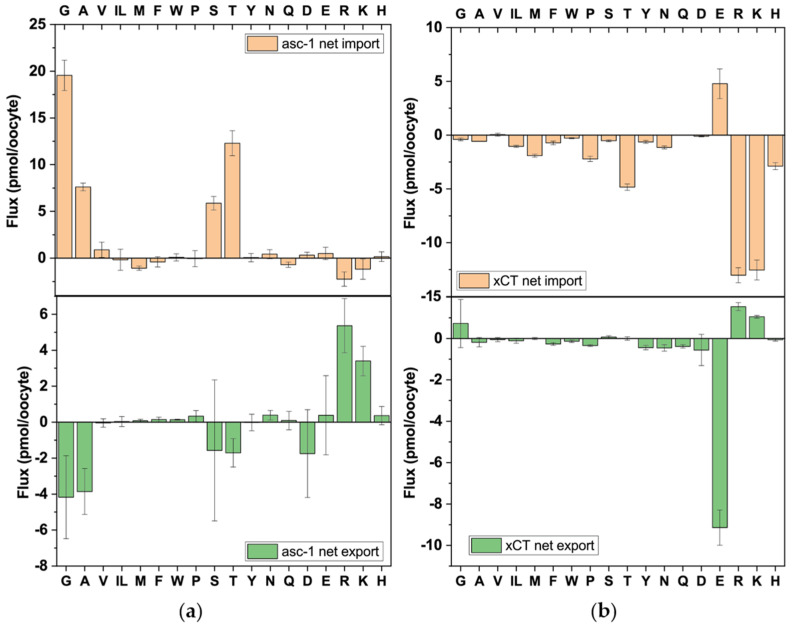
Analysis of antiport via asc-1 and xCT. Oocytes expressing 4F2hc–asc-1 (**a**) or 4F2hc–xCT (**b**) were incubated with labelled and unlabelled amino acids as described in Figure 2a. Net import (positive numbers) and net export (negative numbers) are shown. Isoleucine and leucine cannot be separated by LC–MS and are shown as IL.

**Figure 7 biomolecules-13-00301-f007:**
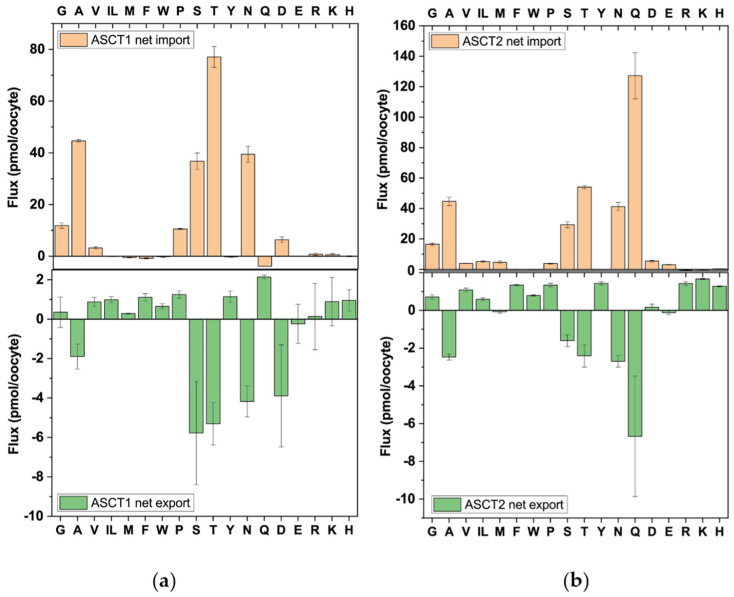
Analysis of antiport via ASCT1 and ASCT2. Oocytes expressing ASCT1 (**a**) or ASCT2 (**b**) were incubated with labelled and unlabelled amino acids as described in Figure 1a. Net import (positive numbers) and net export (negative numbers) are shown. Isoleucine and leucine cannot be separated by LC–MS and are shown as IL.

**Figure 8 biomolecules-13-00301-f008:**
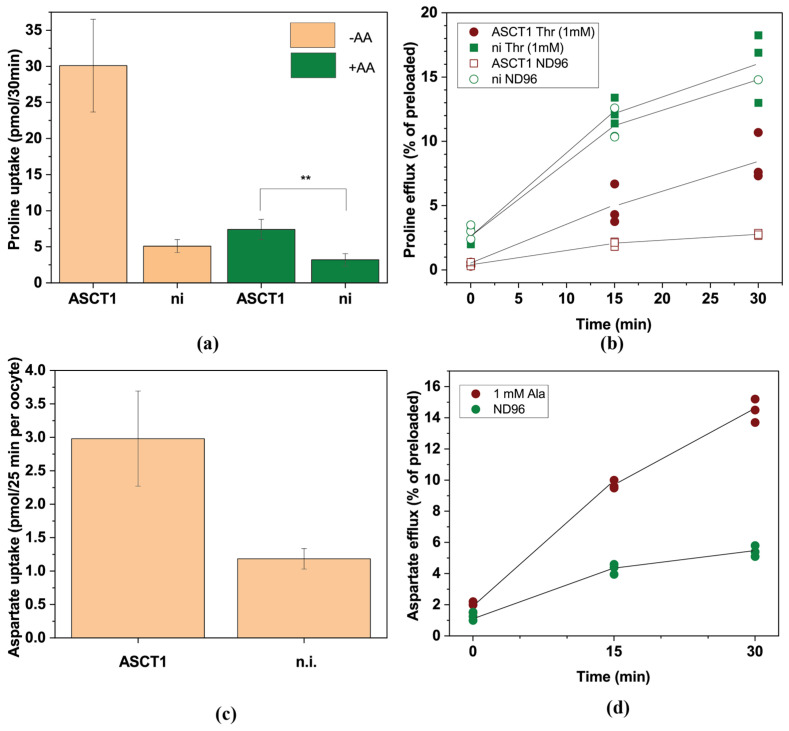
Analysis of ASCT1-mediated proline and aspartate transport. (**a**) Uptake of 0.1 mM [^14^C]proline was measured after 30 min in the presence and absence of an unlabelled amino acid mix (0.1 mM each) in oocytes expressing ASCT1 or in non-injected oocytes. (**b**) Oocytes expressing ASCT1 or non-injected oocytes were preincubated with 0.1 mM [^14^C]proline for 1h. After washing, oocytes were incubated in ND96 in the presence (+Thr) and absence (−Thr) of 1 mM threonine and supernatant samples were analysed for radiolabelled proline after 15 and 30 min. Due to the difference in preloading, efflux is shown as % of preloaded [^14^C]proline. (**c**) Uptake of 0.025 mM [^14^C]aspartate was measured after 25 min in oocytes expressing ASCT1 or in non-injected oocytes. (**d**) Oocytes expressing ASCT1 were preincubated with 0.025 mM [^14^C]aspartate for 2h. After washing, oocytes were incubated in ND96 in the presence (+Ala) and absence (ND96) of 1 mM alanine and supernatant samples were analysed for radiolabelled aspartate after 15 and 30 min. Due to the difference in preloading, efflux is shown as % of preloaded [^14^C]aspartate.

**Table 1 biomolecules-13-00301-t001:** Primers used for Gibson assembly. Lower-case letters denote bases homologous with pGHJ cleaved at the EcoRI site and upper-case letters complement the start and end of the coding sequence of the transporter. Where applicable, the Kozak sequence was designed to match with the optimal consensus sequence.

Transporter	Primer	Sequence (5′–3′)
LAT1	Forward	atcaattccccggggatccgCCACCATGGCGGGTGCGG
	Reverse	agatcaagcttgctctagagCTATGTCTCCTGGGGGACCACC
LAT2	Forward	atcaattccccggggatccgCCACCATGGAAGAAGGAGCCAGG
	Reverse	agatcaagcttgctctagagTCAGGGCTGGGGCTGCCC
y^+^LAT1	Forward	atcaattccccggggatccgCCACCATGGTTGACAGCAC
	Reverse	agatcaagcttgctctagagTTAGTTAGATTTGGGATCCCGTTG
asc-1	Forward	ttccccggggatccgCCACCATGGCCGGCCACACGCA
	Reverse	tcaagcttgctctagagTCATTGTGGCTTCGAGGGCTTG
xCT	Forward	ttccccggggatccgCCACCATGGTCAGAAAGCCTGTT
	Reverse	tcaagcttgctctagagTCATAACTTATCTTCTTCTGGTAC

## Data Availability

The raw data associated with this study are available via Mendeley data at [68].

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
