# Peer review of "Do Amino Acid Antiporters Have Asymmetric Substrate Specificity?"

_biomolecules, 2023, doi:10.3390/biom13020301_

Round 1

Reviewer 1 Report

This interesting manuscript examines whether amino acid antiporters exhibit different substrate specificity in their inward and outward facing conformations.  While these antiporters have been studied in detail, their amino acid specificities have not been examined in more physiological mixtures of all 20 amino acids encoded in DNA.  Studies reported in the present manuscript include the latter design.  Please address the following comments and suggestions.

1.       Line 6 and elsewhere.  While many antiporters catalyze 1:1 exchange, the b0,+AT clearly does not always do so (e.g., Coady et al., 1996, J Membr Biol 149: 1-8).  The authors have made similar observations for the stoichiometry of ASCT2 transport (e.g., their reference 32).  Please consider these observations in your Introduction and Discussion.

2.       Line 33 and elsewhere.  In their reference 1, the concept of biological Maxwell’s Demons seems to deal more with information in enzymes allowing them selectively to catalyze specific biochemical reactions, not the breaking of thermodynamic laws in such reactions.  In parallel, such Demons of biomembrane transport would be information allowing transporters to be selective for their substrates, not the generation of substrate gradients.  Please expand on your thoughts about such selectivity as examples of Maxwell’s Demons, and explain why you believe this concept needs to be modified to include asymmetric transport by obligate amino acid antitporters. 

3.       Lines 33-36. Regarding the above considerations, Van Winkle’s notion, of asymmetric transporters that may have the ability to produce chemical potential solute gradients, seems to deal more with free energy input to produce the asymmetric transporters and their asymmetric membrane environment than with their substrate selectivity as possible Maxwell’s Demons.  This point is emphasized by Van Winkle in footnote 5 on page 89 of his book (reference 2 of the authors).  As far as this reviewer can tell, Van Winkle thinks this free energy input might allow formation of solute gradients by some transporters, but he does not propose such gradients can be formed without free energy input.  Please make it clearer why you think production of such substrate gradients requires either Maxwell’s demons or direct free energy input.

4.       If transporter catalysis of asymmetric flux of a solute across a biomembrane is evidence of the Maxwell demon the authors envision, then we should consider a somewhat simpler example.  Van Winkle’s point appears to be that free energy input does not need to be a direct part of the transport process itself to make the flux asymmetric.  As mentioned in his example of glucose transport across the avian red cell membrane on page 122 of his book, GLUT1 mediated sugar transport switches from antiport to uniport in the presence of mitochondrial inhibitors, possibly owing to covalent modification of GLUT1.  The resultant glucose flux is then asymmetric, but it does not appear to require direct free energy input during transport to make the flux asymmetric.  Perhaps the asymmetric glutamine flux catalyzed by LAT2 and shown in Figure 3 of the manuscript also has an explanation not involving direct free energy input during glutamine transport itself.  In any case, it seems, to this reviewer, likely unnecessary to resort to explanations involving a Maxwell’s Demon when asymmetric transport does not involve direct free energy input.  Please explain more completely how this apparently asymmetric transport for glutamine and other amino acids in your study might be examined further to determine the mechanisms of these phenomena. 

5.       Line 322.  What do the authors predict would be the result of such flux analysis experiments for b0,+AT?  How would these results change for the rBAT(R365W) mutation you have studied?  When expressed in HeLa cells, this mutation seems to selectively abolish arginine efflux by the transporter, and arginine efflux is also defective when the mutant rBAT is expressed in oocytes (Pineda et al., 2004, Biochem J, 377, 665-674).  Thus, the mutation should alter the import and export of amino acids by b0,+AT.

6.       Line 358.  Can this likely explanation be tested?

7.       Lines 414-436. The “overcompensation of arginine and lysine transport” employed by the authors to produce Figure 6 seems especially large (even larger than the measured import and export of the transporters’ amino acid substrates).  Can this “compensation” be calculated more accurately?

8.       Line 426 and Figure 6b.  The glutamate export shown in Figure 6b seems disproportionately large even with the addition of cysteine import, especially when one considers that measured import is much larger than measured export by other antiporters, such as LAT1 and LAT2, and as shown in Figure 3.  How do the authors explain such differences between data shown in Figures 3 and 6b?

9.       Lines 437-466. The stoichiometry of Na+/AA cotransport by system ASC is unusually small for proline relative to most other zwitterionic amino acids (summarized by Van Winke, 2001, Amino Acids, 20, 105-111).  Might this observation help the authors explain the unexpected results for proline discussed in lines 457-466?

Author Response

  1.      Line 6 and elsewhere.  While many antiporters catalyze 1:1 exchange, the b0,+AT clearly does not always do so (e.g., Coady et al., 1996, J Membr Biol 149: 1-8).  The authors have made similar observations for the stoichiometry of ASCT2 transport (e.g., their reference 32).  Please consider these observations in your Introduction and Discussion.

    Response: Many thanks for this comment, we have also added a study from Indiveri’s laboratory, which shows essentially the same behavior as observed by Coady et al. for ASCT2. Our own observation of ASCT2 refers to the exchange of Na+, which may be related to the observations of Coady and Scalise. In any case these studies are introduced and discussed now in more detail.
  2. Line 33 and elsewhere.  In their reference 1, the concept of biological Maxwell’s Demons seems to deal more with information in enzymes allowing them selectively to catalyze specific biochemical reactions, not the breaking of thermodynamic laws in such reactions.  In parallel, such Demons of biomembrane transport would be information allowing transporters to be selective for their substrates, not the generation of substrate gradients.  Please expand on your thoughts about such selectivity as examples of Maxwell’s Demons, and explain why you believe this concept needs to be modified to include asymmetric transport by obligate amino acid antitporters. 

      Response: We have clarified the concept and resulting problems by rewriting the introduction.

  1. Lines 33-36. Regarding the above considerations, Van Winkle’s notion, of asymmetric transporters that may have the ability to produce chemical potential solute gradients, seems to deal more with free energy input to produce the asymmetric transporters and their asymmetric membrane environment than with their substrate selectivity as possible Maxwell’s Demons.  This point is emphasized by Van Winkle in footnote 5 on page 89 of his book (reference 2 of the authors).  As far as this reviewer can tell, Van Winkle thinks this free energy input might allow formation of solute gradients by some transporters, but he does not propose such gradients can be formed without free energy input.  Please make it clearer why you think production of such substrate gradients requires either Maxwell’s demons or direct free energy input.

      Response: Many thanks for this comment. It was our intention to describe Van Winkle’s idea as being consistent with thermodynamic principles but as suggested we may have not been clear enough. We have revised the introduction accordingly.

  1. If transporter catalysis of asymmetric flux of a solute across a biomembrane is evidence of the Maxwell demon the authors envision, then we should consider a somewhat simpler example.  Van Winkle’s point appears to be that free energy input does not need to be a direct part of the transport process itself to make the flux asymmetric.  As mentioned in his example of glucose transport across the avian red cell membrane on page 122 of his book, GLUT1 mediated sugar transport switches from antiport to uniport in the presence of mitochondrial inhibitors, possibly owing to covalent modification of GLUT1.  The resultant glucose flux is then asymmetric, but it does not appear to require direct free energy input during transport to make the flux asymmetric.  Perhaps the asymmetric glutamine flux catalyzed by LAT2 and shown in Figure 3 of the manuscript also has an explanation not involving direct free energy input during glutamine transport itself.  In any case, it seems, to this reviewer, likely unnecessary to resort to explanations involving a Maxwell’s Demon when asymmetric transport does not involve direct free energy input.  Please explain more completely how this apparently asymmetric transport for glutamine and other amino acids in your study might be examined further to determine the mechanisms of these phenomena.

      Response: The discussion has been completely rewritten focusing on mechanistic and structural aspects of antiporter biology.

  1. Line 322.  What do the authors predict would be the result of such flux analysis experiments for b0,+AT?  How would these results change for the rBAT(R365W) mutation you have studied?  When expressed in HeLa cells, this mutation seems to selectively abolish arginine efflux by the transporter, and arginine efflux is also defective when the mutant rBAT is expressed in oocytes (Pineda et al., 2004, Biochem J, 377, 665-674).  Thus, the mutation should alter the import and export of amino acids by b0,+AT.

      Response: This would indeed be the prediction. Notably, the membrane potential already introduces asymmetry in this antiport mechanism. Import of arginine is favoured compared to its export. Because the nature of the endogenous light-chain has not explored systematically, we have not used rBAT in this study.

  1. Line 358.  Can this likely explanation be tested?

      Response: Potentially by injection of large concentrations of 12C amino acids to suppress competition. We have suggested this experiment in the discussion but it is not without problems. Oocytes often leak amino acids from the site of injection, which would spoil the whole experiment because oocytes are pooled in each experiment.

  1. Lines 414-436. The “overcompensation of arginine and lysine transport” employed by the authors to produce Figure 6 seems especially large (even larger than the measured import and export of the transporters’ amino acid substrates).  Can this “compensation” be calculated more accurately?

      Response: This would be difficult, as the competition between endogenous and heterologous light-chain appears to change from batch-to-batch.

  1. Line 426 and Figure 6b.  The glutamate export shown in Figure 6b seems disproportionately large even with the addition of cysteine import, especially when one considers that measured import is much larger than measured export by other antiporters, such as LAT1 and LAT2, and as shown in Figure 3.  How do the authors explain such differences between data shown in Figures 3 and 6b?

    Response: We are not sure what the reviewer is referring to. xCT largely carries out glutamate-glutamate exchange in this experiment. The fluxes are 9 pmol vs 5 pmol (+1 for cystine), which is a reasonable match. In fact it supports the competition argument, which occurs less in the case of glutamate because it is present in mM concentration in oocytes.
  2. Lines 437-466. The stoichiometry of Na+/AA cotransport by system ASC is unusually small for proline relative to most other zwitterionic amino acids (summarized by Van Winkle, 2001, Amino Acids, 20, 105-111).  Might this observation help the authors explain the unexpected results for proline discussed in lines 457-466?

      Response: Many thanks for this comment. The discussion has been completely rewritten including aspects of the ion and substrate exchange by ASC-type transporters.

Reviewer 2 Report

The original paper titled Are Amino Acid Antiporters Biological Maxwell Demons? explores whether aminoacid antiporters substrate specificity can be different for the inward and outward-facing conformation it raises an attractive alternative to analyze fluxes of all known transporter substrates at the same time using mass spectrometry. However, a few details can be improved when displaying the results obtained.

The title of the article is misleading, it suggests a literature review rather than an original paper with experimental findings; it would be convenient to propose a title more restricted to the experimental results.

In the introduction section, the aim of the work including the main purpose and its significance should be explained in a more comprehensive manner framing it in the context of the experimental design and the results obtained.

Figure 1 from showing the computational simulation can be represented with a clearer figure in which all the analyzed amino acids can be distinguished.

Conclusions including the limitations of the proposed method and the experimental findings should be formulated more clearly.

Author Response

The title of the article is misleading, it suggests a literature review rather than an original paper with experimental findings; it would be convenient to propose a title more restricted to the experimental results.

Response: As suggested, we have introduced a new title that reflects the experimental results.

In the introduction section, the aim of the work including the main purpose and its significance should be explained in a more comprehensive manner framing it in the context of the experimental design and the results obtained.

Response: We have completely rewritten the introduction emphasizing mechanistic details of antiporter function.

Figure 1 from showing the computational simulation can be represented with a clearer figure in which all the analyzed amino acids can be distinguished.

Response: As suggested, we have replaced Figure 1 and replaced it with a figure that matches the reported experimental data.

Conclusions including the limitations of the proposed method and the experimental findings should be formulated more clearly.

Response: We have completely rewritten the discussion, concluding with a section dealing with the limitations of the method.

Reviewer 3 Report

The manuscript is extremely interesting and well organized. It deserves publication.

Author Response

No response required

Round 2

Reviewer 1 Report

The authors addressed well all of my comments and suggestions.